# Ageism and the Factors Affecting Ageism among Korean Nursing Students: A Cross-Sectional Study

**DOI:** 10.3390/ijerph18041798

**Published:** 2021-02-12

**Authors:** Jiyeon Ha, Juah Kim

**Affiliations:** 1College of Nursing, Konyang University, Daejeon 35365, Korea; jyhaha403@gmail.com; 2Department of Nursing, Korean Bible University, Seoul 01757, Korea

**Keywords:** aged, ageism, aging, anxiety, interpersonal relations, students, nursing

## Abstract

With the increasing population of older adults, ageism is an obstacle to health equity and can negatively affect older adults’ quality of life and nursing care. This study aims to examine the level of ageism and the factors associated with ageism among nursing college students, who will become the main workforce for gerontological nursing. A cross-sectional survey was conducted among 238 nursing students in two nursing colleges in South Korea. The total score for ageism was 37.06 (SD 5.94) out of a maximum of 72. In the multiple regression model, the predictors of ageism were aging anxiety (*β* = 0.420, *p* < 0.001), frequency of contact (*β* = −0.204, *p* < 0.001), preference for gerontological nursing (*β* = 0.150, *p* = 0.003), age (*β* = 0.145, *p* = 0.003), and quality of contact (*β* = −0.143, *p* = 0.030), revealing that these were factors influencing ageism in the evaluated nursing students. The results suggest that tailored gerontological education programs or community link programs in the nursing curriculum are necessary to share feelings of contact, increase positive experiences with older adults, and reduce anxiety about aging.

## 1. Introduction

As people’s general life expectancy increases, the population of older adults is growing worldwide, and society is aging at a rapid rate. Among Organization for Economic Cooperation and Development countries, it was documented that more than 20% of the population is aged 65 or older in Germany (21.2%), Greece (21.5%), Finland (20.9%), and Italy (22.3%) [1]. In 2020, with the rapidly aging population, the percentage of people aged older than 65 exceeded 15% in South Korea [2]. As the older population grows, social and cultural issues related to older adults are emerging in many countries. Among them, ageism is becoming a considerable social problem that may adversely affect society [3].

Butler [4] first defined ageism as displaying negative attitudes or behaviors toward older adults as they get older. This definition can be used as a reference to prejudice and discrimination against older adults [5]. In a study on ageism conducted in Australia, experiences of ageism for those aged over 60 showed relationships with stress, depressive, and anxious symptoms [6]. About half of Korean older people were reported to have experienced at least one instance of age discrimination [7]. There is a negative perception and prejudice against older adults, as they are portrayed as people in need of support [8] due to age-related incapacity and a high prevalence of chronic diseases [9]; therefore, people may perceive older adults as a category of people that are “weak” and “difficult to treat” in a hospital or community setting. Allen [10] indicates that chronic illnesses in older adults are risk factors for ageism, and that the worse their health condition is, the higher the level of experiencing ageism becomes. Ageism can affect older adults negatively, leading to depression, loneliness, worsening of chronic diseases, and a decline in their subjective health status. Ageism has also been shown to be related to an increased number of older adults experiencing suicidal ideation and poor quality of life [8].

According to a previous study among university students, the more negative their attitudes and behaviors toward older adults are, and the lower the frequency and quality of contact with older adults, the higher their level of ageism becomes [11]. In conditions where young people have more frequent contact or better experiences with older adults, the level of ageism tends to decrease [11]. Barnett and Adams [12] reported that the quality of contact with older adults is associated with ageism, which could be a determinant of aging anxiety. Aging anxiety refers to an individual’s fear of change as they age [13]. Aging anxiety is about oneself, while ageism is directed toward another specific group—that is, older people; hence, it is a distinct concept [12]. However, previous studies have reported that the two concepts are positively associated [14,15,16]. Younger adults have reported high anxiety about aging, and anxiety about aging has been shown to increase bias toward older adults [17,18].

Ageism against older adults also exists among healthcare professionals [19]. It has been reported that about 73% of adults over 65 years old who have multiple chronic diseases have an average of 2.7 chronic diseases [20], meaning that older adults are often in contact with healthcare providers. Nurses are the healthcare professionals who have the most contact with older adults in clinical settings, and older adults often require long-term care and healthcare services; however, negative prejudices against older adults have been found to be more severe than positive biases [21,22]. The younger generation, who has not had much contact with older adults, may lack an understanding of older adults, and this biased view of older adults in a healthcare environment can have a negative impact on quality of nursing care [19,23]. In one study conducted in Australia, most undergraduate nursing students had a higher prevalence of ageist behavior [24]. Thus, nursing students who are future health professionals and educators should pay attention to this issue. In addition, attitudes toward older adults have been found to vary across socio-cultural backgrounds and times [25,26]. A recent study has shown that attitudes toward older adults are more negative in Eastern Asia than in the West [25]. Therefore, this study investigates the level of ageism and factors affecting ageism, particularly contact quality and frequency with older adults as well as anxiety about aging among Korean nursing students in a rapidly aging society.

## 2. Materials and Methods

### 2.1. Study Design

This is a cross-sectional study that employed a convenience sampling method to identify factors associated with ageism, contact experience with older adults, and anxiety about aging among Korean nursing students.

### 2.2. Participants

The participants were nursing students who wished to voluntarily participate in the research. This study was conducted from September to October 2019 at two nursing colleges in Seoul and Daejeon, South Korea. The required sample size for the study was calculated using G*power 3.1 software. The required number of participants was calculated to be 204, considering 0.05 type I error, 0.95 power, 0.15 effect size, and 16 predictor variables in the linear multiple regression analysis. We enrolled 245 participants, considering a 20% dropout rate. Finally, 238 participants’ data were used for the statistical analysis, with seven participants being excluded due to incomplete answers. Among the total study participants, 56 (23.5%) students were in the first year of nursing college, 59 (24.8%) were in the second year, 74 (31.1%) were in the third year, and 49 (20.6%) were in the fourth year.

### 2.3. Measurements

The self-reported survey included questions about sociodemographic and geriatric-related characteristics, ageism, quality of contact, frequency of contact, and aging anxiety.

#### 2.3.1. Sociodemographic and Geriatric-Related Characteristics

The sociodemographic characteristics of participants included age, gender, year in school, religion, marital status, and area of residency while growing up. The geriatric-related characteristics of participants included questions on their experience in gerontological education and nursing practice, residential experience and satisfaction, current residence status and volunteer experience with older adults, and preference for gerontological nursing.

#### 2.3.2. Fraboni Scale of Ageism (FSA)

To measure ageism, we used the FSA developed by Fraboni, Saltstone, and Hughes [27] and translated into Korean by Kim et al. [11]. The Korean version of this measurement consists of 18 items: 7 items for emotional avoidance, 5 items for discrimination, and 6 items for stereotypes. It is scored on a 4-point Likert scale from 1 (strongly disagree) to 4 (strongly agree). The total score range is 18–72; the higher the total, the more ageism toward older adults. The coefficient alpha (or Cronbach’s α) was 0.86 (subscales 0.65–0.77) at development [27] and 0.82 (subscales 0.60–0.84) in the Korean version [11]. This study showed a coefficient alpha value of 0.81 (subscales 0.51–0.83).

#### 2.3.3. Quality of Contact

We used the contact quality with older adults scale developed by Hutchison, Fox, Laas, Matharu, and Urzi [28] and translated by Seo [29] into Korean. This measurement consists of three items. Each item was rated on a 7-point Likert scale from 1 (not at all) to 7 (very well). The total score range is 3–21; the higher the quality score, the more positive the contact. The coefficient alpha value was 0.68 at development [28] and 0.76 for the Korean version [29]. This study showed a coefficient alpha value of 0.71.

#### 2.3.4. Frequency of Contact

We used the contact frequency with older adults scale developed by Hutchison, Fox, Laas, Matharu, and Urzi [28] and translated by Seo [29] into Korean. This three-item questionnaire was rated on a 7-point Likert scale. The total score range is 3–21; the higher the contact frequency score, the more frequent the contact. The coefficient alpha value was 0.62 at development [28] and 0.74 for the Korean version [29]. This study showed a coefficient alpha value of 0.64.

#### 2.3.5. Anxiety about Aging Scale (AAS)

In our study, the AAS, developed by Laser and Faulkender [30] and translated by Kim [31], was used to measure anxiety about aging. This measurement consists of 15 items and includes questions on fear of older adults (five items), psychological concerns (four items), and fear of loss (six items). This tool is intended to indicate how the respondent feels about aging and is rated on a 5-point Likert scale from 1 (strongly agree) to 5 (strongly disagree). The total score range is 15–75; the higher the score, the higher the respondent’s aging anxiety. The coefficient alpha was 0.82 (subscales 0.69–0.78) when developed [30] and 0.86 (subscales 0.74–0.87) for the Korean version [31]. This study showed a coefficient alpha value of 0.83 (subscales 0.67–0.86).

### 2.4. Data Collection

Data were collected from September to October 2019 at two nursing colleges in Seoul and Daejeon, South Korea. Eligible nursing students received self-administered questionnaires that were completed by each participant. Completing the questionnaires took about 20 minutes. Questionnaires were distributed by trained researchers not directly involved in this study. After completion, the questionnaires were gathered using collection boxes in the school.

### 2.5. Ethical Considerations

This study was performed in accordance with the principles expressed in the Declaration of Helsinki and approved by the Institutional Review Board of Konyang University in South Korea (protocol code KYU-2019-278-01). We explained the aims and methods of this study to the participants and confirmed that participation was voluntary. All participants provided written informed consent. After the survey was completed, researchers provided collection boxes for gathering questionnaires, which means that the study was conducted without direct contact between the participants and the researchers.

### 2.6. Data Analysis

The SPSS IBM Statistics Program 22.0 was used to analyze the data, and the statistical significance was set at *p* < 0.05. The sociodemographic and geriatric-related characteristics of the participants were analyzed as descriptive statistics of numbers, percentages, means, and standard deviations (SD). The Shapiro–Wilk test was conducted to test the normality of differences in the mean of ageism according to the characteristics of the participants. A Mann–Whitney U-test and Kruskal–Wallis test were conducted to analyze nonparametric variables, and parametric analysis variables were used for normality by one-way ANOVA. A post-hoc analysis used the Scheffe test. Correlations between variables were analyzed with Pearson correlation coefficient. We used stepwise multiple linear regressions to examine predictors of ageism. To avoid invalid statistical results, we checked multicollinearity by calculating tolerance (0.53–0.98 above 0.2) and variance inflation factors (VIFs, 1.02–1.88 below 5) in the regression model. The results indicate that multicollinearity was not an issue in this study.

## 3. Results

The 238 participants were aged from 18 to 29 with a mean age of 20.7 ± 1.8. Most participants were female (*n* = 215, 90.3%), and 98.3% of the respondents were single. More than half of the participants (*n* = 133, 55.9%) grew up in a metropolitan city. Among participants, 128 (53.8%) had never been educated in gerontological nursing, and 69.3% had no experience in gerontological nursing. Approximately half of the participants (*n* = 123, 51.7%) had no experience living with older adults, and 89.9% of the students were not living with older people at the time of the study. Referring to their preference for gerontological nursing, the answers were “It does not matter” 132 (55.5%), “Yes” 74 (31.1%), or “No” 32 (13.4%) (Table 1). Further analysis of the characteristics of groups who did not prefer gerontological nursing (*n* = 32) showed that 84.4% of them did not have experience in gerontological nursing (*n* = 27), and 65.5% of the respondents did not have experience of living with older adults (*n* = 21).

The descriptive statistics of the variables used in this study are as shown in Table 2. The total score for ageism was 37.06 (SD 5.94) out of a maximum of 72. The participants’ contact quality score with older adults was 11.25 (SD 3.26), and the contact frequency score was 13.40 (SD 4.02) out of a maximum of 21. The aging anxiety score was 42.43 (SD 7.71) out of a maximum of 75.

Among the sociodemographic characteristics, year in school (F = 4.414, *p* = 0.005) and area of residency while growing up (F = 4.186, *p* = 0.016) showed a significant difference in ageism. The results for preference for gerontological nursing (F = 20.945, *p* < 0.001) were significantly different from those for ageism on geriatric-related characteristics (Table 3).

Contact quality (r = −0.486, *p* < 0.001) and contact frequency (r = −0.449, *p* < 0.001) had a negative correlation with ageism. Aging anxiety (r = 0.581, *p* < 0.001) had a positive correlation with ageism (Table 4).

To identify factors affecting ageism among the participants, a stepwise regression analysis was performed by adding variables that are correlated with ageism, including quality of contact, frequency of contact, aging anxiety, age, year in school, area of residency while growing up, and preferences for gerontological nursing. The result of the regression model for the participants’ ageism was significant (F = 41.775, *p* < 0.001), and the predictors of ageism were aging anxiety (*β* = 0.420, *p* < 0.001), frequency of contact (*β* = −0.204, *p* < 0.001), preference for gerontological nursing (*β* = 0.150, *p* = 0.003), age (*β* = 0.145, *p* = 0.003), and quality of contact (*β* = −0.143, *p* = 0.030). The explanatory power of the variables for ageism was 46.2% (Table 5).

## 4. Discussion

The purpose of this study was to determine the level of ageism and the factors affecting ageism among Korean nursing students. The total score for ageism was 37.06 out of a maximum of 72 among Korean nursing students, indicating that Korean nursing students’ ageism toward older adults is neutral. A recent study showed that the score for ageism was 39.75 out of 72 points among Korean nurses in general hospitals, which indicates that the ageism of Korean nursing students was slightly lower than Korean nurses’ ageism [16]. Some previous studies on nurses’ ageism showed similar results to our study [32,33], whereas a previous study of 18–25-year-old adults in Turkey showed that their attitudes toward older adults were generally positive [34]. Attitudes toward older adults have also been shown to differ based on social background and ethnicity within the Asian culture [25]. In general, Eastern Asian cultures are known to have a higher respect for older adults than Western culture [25]. Ageism is a concept interpreted within the social and cultural context [35]; this result should take the Korean socio-cultural environment into account, specifically the rapid increase in the older adults population [2].

In this study, the analysis of ageism’s predictors in nursing students explained 46.2% of the variance. These predictors included aging anxiety, quality and frequency of contact, preference for gerontological nursing, and age. Anxiety about aging can lead to stereotyping and a biased attitude toward older people [17]. The younger, more inexperienced, and less educated healthcare professionals are, the more anxious they are about aging [18]. In addition, young students can be less perceptive regarding older adults, due to having fewer experiences with them and less knowledge about them [36]. Therefore, gerontological content during undergraduate programs is required for nursing students to recognize aging as a natural course of life and to help them acquire accurate knowledge of aging. It is also necessary to teach these students to share the feelings about older adults that they may experience in clinical settings and to seek ways to reduce aging anxiety using the gerontological nursing education curriculum.

Our results showed that the quality and frequency of contact experiences with older adults were associated with ageism. This result is similar to that of a previous study, which stated that increased contact or positive contact with older adults tends to reduce ageism among university students [11]. However, the frequency of contact was an inconsistent factor of ageism. Drury and colleagues [37] suggest that having frequent contact with older adults may not be sufficient, but more relevant contact quality reduces ageism. Levy [38] also stressed that personal experience of positive contact with older adults is helpful in reducing ageism. As the number of nuclear families has been increasing recently, the younger generation does not have much contact experience with older adults, and their awareness of older adults may be fragmentary or lacking [36]. The contact experience of nursing students may be biased, as the older adults that they come into contact with in clinical settings are often vulnerable or have health problems [39]. A study on nursing students in Australia showed that little contact with older adults and a lack of previous experience were barriers to working in gerontological care [40]. Nurses are healthcare providers who manage health-related problems and quality of life for older adults in clinical settings [21]. Nursing students need to engage with gerontological experiences in the classroom or in clinical settings to improve their attitudes toward older people [41]. Tailored education programs or community link programs for nursing students can enhance positive contact experiences with older adults.

Our current study shows that the preference for gerontological nursing is a factor of ageism. This confirms the findings of a previous study, which stated that positive bias is a factor in the intention of working in a geriatric setting [40]. A higher preference for working with older adults is associated with positive attitudes toward older adults [17,42]. However, some studies reported that undergraduate students were reluctant to choose gerontological care, strengthening the negative image of clinical experience with older adults [43]. In the present study, among those who do not prefer gerontological nursing care, 84.4% had no experience in gerontological care. Hovey et al. [41] indicated that the more nursing education students received, the more positive their attitudes toward older adults. Stevens [43] emphasized that ageism and negative clinical experiences in a community affect nursing students’ career choice. As this negative perception and attitude toward older adults negatively affects their health outcomes [10], it is necessary to develop a variety of content to increase the preference for gerontological nursing care and to provide a simulation practice environment with older patients in a standardized nursing curriculum.

Our results also indicate that the age of nursing students was one of the factors of ageism. In particular, third year students had a higher score of ageism than first year students, which can be inferred as the result of increased contact with sick older people in hospital clinical practice. Inconsistent results have been found with respect to the association between age and ageism [43,44,45]. Donizzetti [45] reported a positive correlation between age and ageism, while Kim and Ha [16] found that age is not associated with ageism. Cultural factors that are rapidly increasing the aging population in South Korea could also affect ageism [35]. Since our study was conducted with nursing students between the ages of 18 and 29, the age range is very limited. Therefore, a large-scale study including nursing students and healthcare providers from different cultures is needed to determine whether age is associated with ageism.

The current study has several limitations. First, this study is limited in the generalizability of the results, as it collected data using a convenience sampling method from only two institutions. Second, while variables were measured based on prior studies to identify the associated factors of ageism, we did not consider the diverse variables of the social and cultural background of nursing students. Third, given the cross-sectional design of the study, causal inferences are limited in relation to ageism, contact experience, and aging anxiety. Last, the majority of participants in this study were female. Future studies should investigate the gender differences of ageism.

## 5. Conclusions

This study found that nursing students’ ageism toward older adults was neutral and the influencing factors of ageism were aging anxiety, quality and frequency of contact, preference for gerontological nursing, and age. As negative perceptions and attitudes toward older people could negatively affect the quality of care and health status of older adults in the community, educators should make a concerted effort to increase positive contact experiences with older people in undergraduate programs. In addition, nursing students, who will be responsible for the quality of care for older adults in the future, need a better understanding of aging; it is important to correct their misconceptions about older adults that lead to aging anxiety before starting work at the hospital. The findings of this study could contribute to an understanding of the importance of positive experiences with older adults and demonstrate the necessity of finding various ways within the nursing curriculum to reduce anxiety about aging in order to improve the quality of public health.

## Figures and Tables

**Table 1 ijerph-18-01798-t001:** Sociodemographic and geriatric-related characteristics of participants (*N* = 238).

Characteristics	Categories	*n* (%)/Mean (SD)
Age	18–29	20.7 (1.8)
Gender	Female	215 (90.3)
	Male	23 (9.7)
Year in school	1	56 (23.5)
	2	59 (24.8)
	3	74 (31.1)
	4	49 (20.6)
Religion	Christianity	120 (50.4)
	Catholicism	12 (5.1)
	Buddhism	5 (2.1)
	None	101 (42.4)
Marital status	Single	234 (98.3)
	Married	4 (1.7)
Area of residency while growing up	Metropolitan city	133 (55.9)
	Small city	68 (28.6)
	Countryside	37 (15.5)
Gerontological education	Yes	110 (46.2)
	No	128 (53.8)
Gerontological nursing practice	Yes	73 (30.7)
	No	165 (69.3)
Residential experience with older adults	Yes	115 (48.3)
	No	123 (51.7)
Residential experience satisfaction	1–5	4.16 (0.96)
Current residence status with older adults	Yes	24 (10.1)
	No	214 (89.9)
Volunteer experience with older adults	Yes	210 (88.2)
	No	28 (11.8)
Preference for gerontological nursing	Yes	74 (31.1)
	No	32 (13.4)
	It does not matter	132 (55.5)

**Table 2 ijerph-18-01798-t002:** Descriptive statistics for ageism, quality and frequency of contact experience, and aging anxiety (*N* = 238).

Variables	Mean (SD)	Range
Ageism	37.06 (5.94)	21–53
Quality of contact	11.25 (3.26)	3–21
Frequency of contact	13.40 (4.02)	4–21
Aging anxiety	42.43 (7.71)	19–71

**Table 3 ijerph-18-01798-t003:** Differences in ageism according to participant characteristics (*N* = 238).

Characteristics	Categories	Ageism	
		Mean (SD)	t/Z/F (*p*)Scheffe’s Test
Gender	Female	2.07 (0.33)	−1.554 (0.060) ^†^
	Male	1.96 (0.32)	
Year in school	1 ^a^	1.99 (0.27)	4.414 (0.005) **
	2 ^b^	2.02 (0.38)	a < c
	3 ^c^	2.17 (0.31)	
	4 ^d^	2.02 (0.32)	
Religion	Christianity	2.04 (0.33)	0.440 (0.725)
	Catholicism	2.11 (0.29)	
	Buddhism	2.09 (0.38)	
	None	2.08 (0.33)	
Marital status	Single	2.06 (0.33)	0.205 (0.511)
	Married	2.17 (0.18)	
Area of residency while growing up	Metropolitan city ^a^	2.07 (0.32)	4.186 (0.016) * ^‡^
	Small city ^b^	2.11 (0.32)	c < b
	Countryside ^c^	1.93 (0.35)	
Gerontological education	Yes	2.10 (0.36)	1.771 (0.078)
	No	2.02 (0.32)	
Gerontological nursing practice	Yes	2.04 (0.34)	0.298 (0.632)
	No	2.07 (0.32)	
Residential experience with older adults	Yes	2.02 (0.34)	0.835 (0.055)
	No	2.10 (0.32)	
Current residence status with older adults	Yes	1.99 (0.37)	0.411 (0.304)
	No	2.07 (0.33)	
Volunteer experience with older adults	Yes	2.04 (0.32)	0.278 (0.066)
	No	2.17 (0.38)	
Preference for gerontological nursing	Yes ^a^	1.91 (0.32)	20.945 (<0.001) **
	No ^b^	2.32 (0.27)	a < c < b
	It does not matter ^c^	2.08 (0.31)	

^†^: Mann–Whitney U-test, ^‡^: Kruskal–Wallis test. * *p* < 0.05, ** *p* < 0.01.

**Table 4 ijerph-18-01798-t004:** Correlations between ageism, quality and frequency of contact experience, and aging anxiety (*N* = 238).

Variables	r (*p*)			
	1	2	3	4
1. Ageism	1			
2. Quality of contact	−0.486 (<0.001) **	1		
3. Frequency of contact	−0.449 (<0.001) **	0.599 (<0.001) **	1	
4. Aging anxiety	0.581 (<0.001) **	−0.475 (<0.001) **	−0.293 (<0.001) **	1

* *p* < 0.05, ** *p* < 0.01.

**Table 5 ijerph-18-01798-t005:** Stepwise multiple regression model for predictors of ageism (*N* = 238).

Predictors	Adjusted R^2^	B	SE	*β*	t	*p*
Aging anxiety	0.462	0.270	0.035	0.420	7.685	<0.001
Frequency of contact		−0.050	0.015	−0.204	−3.400	<0.001
Preference for gerontological nursing (no)		0.145	0.048	0.150	3.043	0.003
Age		0.027	0.009	0.145	3.009	0.003
Quality of contact		−0.043	0.020	−0.143	−2.183	0.030
F (*p*) = 41.775 (<0.001)						

B: unstandardized estimates. SE: standardized error. *β*: standardized estimates.

## Data Availability

The data presented in this study are available on request from the corresponding author. The data are not publicly available due to privacy or ethical restrictions.

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
