# Peer review of "Ageism and the Factors Affecting Ageism among Korean Nursing Students: A Cross-Sectional Study"

_ijerph, 2021, doi:10.3390/ijerph18041798_

Round 1
Reviewer 1 Report
Overall paper is well written, uses nice dataset and reaches clear and logical results. My minor comments are as follows:
- I will ask authors to provide to summarize the main findings in bulleted format and put in a new section as clinical perspectives.
- During which months/years Overall paper is well written, uses nice dataset and reaches clear and logical results. was the study conducted?
- In the discussion, the authors should better describe the results, limitations and compare with other research in the area.
- As a result of some of the previous comments, specially the scope of this paper, the conclusions do not seem to be very relevant. What does this article contribute to scientific knowledge? What are the news? Why are the results of this article useful?
In the discussion or the conclusions section, these questions should be answered.
Author Response
Overall paper is well written, uses nice dataset and reaches clear and logical results. My minor comments are as follows:
Point 1: I will ask authors to provide to summarize the main findings in bulleted format and put in a new section as clinical perspectives.
Response 1: Thank you for your suggestion. In accordance with the journal author guidelines, the main findings of this study were emphasized in the conclusion section.
“This study found that nursing students’ ageism toward older adults are neutral and the influencing factors of ageism were aging anxiety, quality and frequency of contact, preference for geriatric nursing, and age. As negative perceptions and attitudes toward older people could negatively affect quality of care and health status of older adults in the community, educators should make a concerted effort to increase positive contact experiences with older people in undergraduate programs. In addition, nursing students, who will be responsible for the quality of care for older adults in the future, need a better understanding of aging; it is important to correct their misconceptions about older adults that lead to aging anxiety before starting work at the hospital. The findings of this study could contribute to understanding the importance of positive experiences with older adults and demonstrating the necessity of finding various ways to reduce anxiety about aging in nursing curriculum to improve the quality of public health.” (page 8 line 296-307)
Point 2: During which months/years was the study conducted?
Response 2: We have added information regarding the data collection period.
“This study was conducted from September to October 2019 at two nursing colleges in Seoul and Daejeon, South Korea.” (page 2, line 84)
Point 3: In the discussion, the authors should better describe the results, limitations and compare with other research in the area.
Response 3: Thank you for your comment. We have revised some sentences in the discussion.
“In addition, young students can be less perceptive regarding older adults due to having fewer experiences and less knowledge about them [36].” (page 7, line 235-236)
“Nurses are healthcare providers who manage health-related problems and quality of life concerns for older adults in clinical settings [41]. Nursing students need to engage with gerontological experiences in the classroom or in clinical settings to improve their attitudes toward older people [42]. Tailored education programs or community link programs for nursing students can be improved through positive contact with older adults.” (page 7-8, line 255-260)
“Cultural factors that are rapidly increasing the aging population in Korea can also affect ageism [35]. Since our study was conducted with nursing students between the ages of 18 and 29, the age range is very limited. Therefore, a large-scale study including nursing students and healthcare providers from different cultures is needed to determine whether age is associated with ageism.” (page 8, line 282-286)
“The current study has several limitations. First, this study is limited in the generalizability of the results, as it collected data using a convenience sampling method from only two institutions. Second, while variables were measured based on prior studies to identify the associated factors of ageism, we did not consider the diverse variables of the social and cultural background of nursing students. Third, given the cross-sectional design of the study, causal inferences are limited in relation to ageism, contact experience, and aging anxiety. Last, the majority of participants in this study were female. Future studies should investigate the gender differences of ageism.” (page 8, line 287-294)
Point 4: As a result of some of the previous comments, specially the scope of this paper, the conclusions do not seem to be very relevant. What does this article contribute to scientific knowledge? What are the news? Why are the results of this article useful?
Response 4: The contributions of this study are presented in the conclusion.
“As negative perceptions and attitudes toward older people could negatively affect quality of care and health status of older adults in the community, educators should make a concerted effort to increase positive contact experiences with older people in undergraduate programs. In addition, nursing students, who will be responsible for the quality of care for older adults in the future, need a better understanding of aging; it is important to correct their misconceptions about older adults that lead to aging anxiety before starting work at the hospital. The findings of this study could contribute to understanding the importance of positive experiences with older adults and demonstrating the necessity of finding various ways to reduce anxiety about aging in nursing curriculum to improve the quality of public health..” (page 8 line 298-307)
Point 5: In the discussion or the conclusions section, these questions should be answered.
Response 5: Thank you for your comment. We have revised accordingly.
“In addition, young students can be less perceptive regarding older adults due to having fewer experiences and less knowledge about them [36].” (page 7, line 235-236)
“Nurses are healthcare providers who manage health-related problems and quality of life concerns for older adults in clinical settings [41]. Nursing students need to engage with gerontological experiences in the classroom or in clinical settings to improve their attitudes toward older people [42]. Tailored education programs or community link programs for nursing students can be improved through positive contact with older adults.” (page 7-8, line 255-260)
“Cultural factors that are rapidly increasing the aging population in Korea can also affect ageism [35]. Since our study was conducted with nursing students between the ages of 18 and 29, the age range is very limited. Therefore, a large-scale study including nursing students and healthcare providers from different cultures is needed to determine whether age is associated with ageism.” (page 8, line 282-286)
Reviewer 2 Report
Thank you for sending your paper entitled “Ageism and the Factors Affecting Ageism among Korean Nursing Students: A Cross-Sectional Study” to Internacional Journal Environmental Research and Public Health. After carefully review this interesting paper, the following comments are listed for your reference:
- Abstract: To increase potential citations, authors should check keywords against those recommended in the MeSH Browser of Medical Subject Headings https://meshb.nlm.nih.gov/search. For example: “contact” is not MeSH. I recommend that you change this keyword.
- Introduction (Lines 31-32): If Butler's definition of ageism is literally copied, you should put it in quotes. If you do not do it, plagiarism is allowed.
- Methods (Lines 80-88): What academic year were the nursing students? Nursing first year students are not the same as third or fourth year ... This must be indicated in the participants section. It can also skew the results if it is not taken into account.
- The results are very well structured.
- The discussion is very well developed.
- References (P9-10): 37/41 are more than 5 years old and 28 of them are more than 10 years old. Referencing recent and relevant literature would make the paper more robust, especially in the introduction and discussion sections. If this is not the case (there is no recent literature), it should be noted appropriately in each section and limitation section. The bibliography is related to the research. In any case, it is recommended that more bibliographical references from the last five years be used.
- References (P9-10): I recommend checking the citations according to the journal's standards.The DOI must be added in each bibliographic reference.
Author Response
Point 1: Abstract: To increase potential citations, authors should check keywords against those recommended in the MeSH Browser of Medical Subject Headings https://meshb.nlm.nih.gov/search. For example: “contact” is not MeSH. I recommend that you change this keyword.
Response 1: Thank you. We have changed ‘contact’ to ‘interpersonal relations’ in the keywords.
“Keywords: aged; ageism; aging; anxiety; interpersonal relations; students; nursing” (page 1 line 21)
Point 2: Introduction (Lines 31-32): If Butler's definition of ageism is literally copied, you should put it in quotes. If you do not do it, plagiarism is allowed.
Response 2: Thank you for your comment. To clarify, we did not directly quote Butler in the text; we did paraphrase the quotation and then reference it. We confirmed that there were no issues related to plagiarism through the plagiarism test program. (https://www.ithenticate.com).
Point 3: Methods (Lines 80-88): What academic year were the nursing students? Nursing first year students are not the same as third or fourth year ... This must be indicated in the participants section. It can also skew the results if it is not taken into account.
Response 3: Thank you for your suggestion. We have addressed a sentence about the participants’ academic year.
“Among the total study participants, 56 (23.5%) students were in the first year of nursing college, 59 (24.8%) were in the second year, 74 (31.1%) were in the third year, and 49 (20.6%) were in the fourth year.” (page 2 line 90-93)
Point 4: References (P9-10): 37/41 are more than 5 years old and 28 of them are more than 10 years old. Referencing recent and relevant literature would make the paper more robust, especially in the introduction and discussion sections. If this is not the case (there is no recent literature), it should be noted appropriately in each section and limitation section. The bibliography is related to the research. In any case, it is recommended that more bibliographical references from the last five years be used.
Response 4: Thank you for your comments. We tried to change out of date references (6, 7, 22, 23, 35, 36, 41, 42) and some sentences are revised.
“In a study on ageism conducted in the Australia, experiences of ageism for those aged over 60 showed relationships with stress, depressive, and anxious symptoms [6]. About half of Korean older people were reported to have experienced at least one instance of age discrimination [7].” (page 1, line 34-37)
“In addition, young students can be less perceptive regarding older adults due to having fewer experiences and less knowledge about them [36].” (page 7, line 235-236)
“Nurses are healthcare providers who manage health-related problems and quality of life concerns for older adults in clinical settings [41]. Nursing students need to engage with gerontological experiences in the classroom or in clinical settings to improve their attitudes toward older people [42]. Tailored education programs or community link programs for nursing students can be improved through positive contact with older adults.” (page 7-8, line 255-260)
Point 5: References (P9-10): I recommend checking the citations according to the journal's standards. The DOI must be added in each bibliographic reference.
Response 5: Thank you. We have added the DOI in each bibliographic reference.
“3. Laganá, L.; Gavrilova, L.; Carter, D.B.; Ainsworth A.T. A Randomized Controlled Study on the Effects of a Documentary on Students' Empathy and Attitudes towards Older Adults. Psychol. Cogn. Sci. 2017, 3, 79-88. doi: 10.17140/PCSOJ-3-127” (page 9-11)
Reviewer 3 Report
This is a study of ageism among nursing students at two Korean nursing schools. The authors submitted a questionnaire to nursing students who volunteered to complete it. They performed a sample size calculation. Data from 238 participants was used. This exceeded the required number of participants.
The authors used an instrument to measure ageism that was translated into Korean. They report the raw coefficient alpha for the instrument (0.86) as well as in their study (0.81).
They also used instruments for quality of contact, frequency of contact, and anxiety about aging. The raw coefficient alpha for these was lower except for anxiety about aging. Note here that Cronbach preferred the term raw coefficient alpha over Cronbach alpha so the authors might want to change this term.
The authors tested for normality of differences in the mean of ageism. This suggests that raw coefficient alpha and Pearson correlation coefficients are appropriate.
They used a Mann-Whitney test and a Kruskal Wallace test to analyze nonparametric variables as well as ANOVA. They evaluated correlations using Pearson and applied stepwise multiple linear regression to examine predictors of ageism. They tested for multicollinearity.
Significant findings include a significant negative correlation between contact quality and contact frequency and ageism as well as a significant positive correlation between aging anxiety and ageism. These are important enough to probably make it into the abstract.
The authors used stepwise regression by adding variables correlated with ageism. There are statistical problems with stepwise regression. However, it is commonly used. So long as the only real reliance is on the final model this does not constitute a problem for this analysis. The model has meaning with an F of 41.775. The adjusted R square is reasonable.
The authors might want to discuss effect size. While frequency of contact and quality of contact have negative coefficients, their beta coefficients are small. The model coefficient for aging anxiety is positive and large and those who do not prefer geriatric nursing had positive and moderately large coefficients.
The authors characterize the adjusted R square as showing a “significant regression.” This probably misstates what the adjusted R square means. Their model explains 42.6% of the variance.
The paper’s discussion includes reference to other work. These findings are generally consistent.
The conclusions are appropriate observations given the findings. They state that “nursing students should work toward a better understanding of aging and reduce their misconceptions about older adults.”
All told, I recommend acceptance of this paper with some minor revisions. From a drafting standpoint I converted the PDF to Word. This produced extra spaces that have come from the conversion or it may be in the typeset manuscript. It’s worth double checking.
Author Response
Point 1: This is a study of ageism among nursing students at two Korean nursing schools. The authors submitted a questionnaire to nursing students who volunteered to complete it. They performed a sample size calculation. Data from 238 participants was used. This exceeded the required number of participants.
Response 1: As written in the text, we planned to collect data from 245 people considering the dropout rate of 20% in addition to the number of participants calculated by G*power software. The number of participants included in the final analysis was 238.
“The required sample size for the study was calculated using G*power software. The required number of participants was calculated to be 204, considering a type I error of 0.05, a power of 0.95, an effect size of 0.15, and 16 predictor variables in the linear multiple regression analysis. We enrolled 245 participants, considering a 20% dropout rate. Finally, 238 participants’ data were used for the statistical analysis, with seven participants being excluded due to incomplete answers.” (page 2 line 85-90)
Point 2: They also used instruments for quality of contact, frequency of contact, and anxiety about aging. The raw coefficient alpha for these was lower except for anxiety about aging. Note here that Cronbach preferred the term raw coefficient alpha over Cronbach alpha so the authors might want to change this term.
Response 2: Thanks for your suggestion. We have revised ‘Cronbach alpha’ to ‘coefficient alpha’ in the methods section.
“The coefficient alpha (or Cronbach’s ⍺) was 0.86 (subscales .65–.77) at development [27] and 0.82 (subscales .60–.84) in the Korean version [11]. This study showed a coefficient alpha value of 0.81 (subscales .51–.83).” (page 3 line 110-112)
“The coefficient alpha value was 0.68 at development [28] and 0.76 for the Korean version [29]. This study showed a coefficient alpha value of 0.71.” (page 3 line 118-119)
“The coefficient alpha value was 0.62 at development [28] and 0.74 for the Korean version [29]. This study showed a coefficient alpha value of 0.64.” (page 3 line 124-126)
“The coefficient alpha was 0.82 (subscales .69–.78) when developed [30] and 0.86 (subscales .74–.87) for the Korean version [31]. This study showed coefficient alpha value of 0.83 (subscales .67–.86).” (page 3 line 134-136)
Point 3: (Discussion line 228) The authors characterize the adjusted R square as showing a “significant regression.” This probably misstates what the adjusted R square means. Their model explains 42.6% of the variance.
Response 3: Thank you for your comment. We deleted the words “significant regression” and revised the sentence.
“In this study, the analysis of ageism’s predictors in nursing students explained 46.2% of the variance.” (page 7, line 230-231)